# Effect of Anti-Osteoporotic Treatments on Circulating and Bone MicroRNA Patterns in Osteopenic ZDF Rats

**DOI:** 10.3390/ijms23126534

**Published:** 2022-06-10

**Authors:** David Carro Vázquez, Lejla Emini, Martina Rauner, Christine Hofbauer, Johannes Grillari, Andreas B. Diendorfer, Richard Eastell, Lorenz C. Hofbauer, Matthias Hackl

**Affiliations:** 1TAmiRNA GmbH, Department of Research, Leberstrasse 20, 1110 Vienna, Austria; david.carro.vazquez@tamirna.com (D.C.V.); andreas.diendorfer@tamirna.com (A.B.D.); 2Center for Healthy Aging and Department of Medicine III, Technische Universität Dresden, 01069 Dresden, Germany; lejla.emini@uniklinikum-dresden.de (L.E.); martina.rauner@uniklinikum-dresden.de (M.R.); christine.hofbauer@ukdd.de (C.H.); lorenz.hofbauer@ukdd.de (L.C.H.); 3Ludwig Boltzmann Institute for Traumatology in Cooperation with AUVA, Ludwig Boltzmann Society, 1200 Vienna, Austria; johannes.grillari@trauma.lbg.ac.at; 4Institute of Molecular Biotechnology, University of Natural Resources and Life Sciences, 1180 Vienna, Austria; 5Austrian Cluster for Tissue Regeneration, 1200 Vienna, Austria; 6Academic Unit of Bone Metabolism and Mellanby Centre for Bone Research, University of Sheffield, Sheffield S10 2RX, UK; r.eastell@sheffield.ac.uk

**Keywords:** microRNA, type 2 diabetes, ZDF, biomarker, next-generation sequencing, osteoporosis, circulating microRNA

## Abstract

Bone fragility is an adverse outcome of type 2 diabetes mellitus (T2DM). The underlying molecular mechanisms have, however, remained largely unknown. MicroRNAs (miRNAs) are short non-coding RNAs that control gene expression in health and disease states. The aim of this study was to investigate the genome-wide regulation of miRNAs in T2DM bone disease by analyzing serum and bone tissue samples from a well-established rat model of T2DM, the Zucker Diabetic Fatty (ZDF) model. We performed small RNA-sequencing analysis to detect dysregulated miRNAs in the serum and ulna bone of the ZDF model under placebo and also under anti-sclerostin, PTH, and insulin treatments. The dysregulated circulating miRNAs were investigated for their cell-type enrichment to identify putative donor cells and were used to construct gene target networks. Our results show that unique sets of miRNAs are dysregulated in the serum (*n* = 12, FDR < 0.2) and bone tissue (*n* = 34, FDR < 0.2) of ZDF rats. Insulin treatment was found to induce a strong dysregulation of circulating miRNAs which are mainly involved in metabolism, thereby restoring seven circulating miRNAs in the ZDF model to normal levels. The effects of anti-sclerostin treatment on serum miRNA levels were weaker, but affected miRNAs were shown to be enriched in bone tissue. PTH treatment did not produce any effect on circulating or bone miRNAs in the ZDF rats. Altogether, this study provides the first comprehensive insights into the dysregulation of bone and serum miRNAs in the context of T2DM and the effect of insulin, PTH, and anti-sclerostin treatments on circulating miRNAs.

## 1. Introduction

Diabetes mellitus has emerged as a novel risk factor for fragility fractures. Type 2 diabetes mellitus (T2DM) accounts for 90% of all diabetes cases in adults and, due to the epidemic of obesity, its prevalence has further gradually increased in recent years. T2DM affects bone health in the later stages of the disease, with less severe bone mass reduction and risk of fracture compared to T1DM [1]. As the diabetes epidemic is increasing worldwide with aging, and the fractures that are associated with diabetes cause an increase in morbidity, mortality, and healthcare costs, diabetes-induced osteoporosis imposes a significant burden on our society and our healthcare system, and efforts are being made to study the underlying molecular mechanisms involved to establish an early-detection method and to identify therapy options for diabetic bone disease [1]. In contrast to T1DM, BMD is not reduced in patients with T2DM, with a normal or even 5–10% higher mean BMD. This indicates that bone fragility in each form of diabetes develops via distinct mechanisms that to date remain largely unknown and may require an individualized approach for effective treatment [1].

In this regard, recent studies suggest a crucial role of miRNAs in diabetes mellitus as well as in bone development and homeostasis [2,3,4]. microRNAs (miRNAs) are small non-coding RNAs that suppress mRNA translation or induce its degradation to regulate the expression of entire gene networks and important cellular processes at a posttranscriptional level thanks to their ability to target up to 100 distinct mRNAs [2,3,4].

Many studies have also shown that miRNAs can be involved in the pathophysiology of osteoporosis and reduced fracture healing in both T1DM and T2DM. Takahara et al. [5] and Tang et al. [6] investigated miRNA regulation in rats with T1DM induced via a streptozotocin injection. Takahara et al. reported that during femur fracture healing, five miRNAs (miR-140-3p, miR-140-5p, miR-181a-1-3p, miR-210-3p, and miR-222-3p) showed changing patterns of expression in the newly generated tissue at the fracture site of the diabetic rat model. Tang et al. showed an upregulation of miR-203-3p, concluding that this miRNA can serve as a putative biomarker for diabetic bone loss and targets Smad1, inhibiting the BMP/Smad pathway and acting in that way as a suppressor of osteogenesis. In their study, Grieco et al. found that miR-21-5p and miR-148a-5p were upregulated in T1DM patients and correlated to markers of bone strength and metabolism, such as total body bone mineral density and circulating PTH [7].

To investigate the utility of circulating miRNAs as biomarkers of T2DM bone disease, we recently evaluated the circulating miRNA profiles of 80 postmenopausal women with and without T2DM [8]. With 48 significantly differentially expressed miRNAs, T2DM patients with fractures exhibited stronger shifts in circulating miRNAs relative to T2DM controls without fractures or the non-diabetic study arm (23 differentially regulated miRNAs). We confirmed these findings in an independent cohort of 168 T2D postmenopausal women [9], showing that specific miRNAs are associated with incident fragility fracture in older diabetic women, concluding that circulating miR-31-5p and miR-203a can be used as prognostic biomarkers for fracture risk in the context of T2D.

In order to further investigate the involvement of specific miRNAs in T2DM bone disease and anti-osteoporotic treatment options, here, we aimed to analyze serum and bone tissue samples from a well-established rat animal model of T2DM, the Zucker Diabetic Fatty (ZDF) model. This model has been previously described by us [10], revealing suppressed osteoblastogenesis as a cause and mechanism for low bone mass and impaired bone regeneration. We [11,12] and others [13] further showed that different treatment options such as anti-sclerostin (anti-scl), parathyroid hormone (PTH), or insulin have specific effects on the improvement of bone quality and bone healing in the context of T2DM using ZDF rats. In this study, we employed an optimized and untargeted microRNA next-generation sequencing (NGS) discovery assay called miND to quantify miRNA levels in serum and ulna samples from the ZDF rat model of T2DM collected in our earlier studies. With the generated NGS data, we performed hierarchical clustering analysis, univariate and multivariate statistical analysis, as well as gene target network analysis and cell-type enrichment analysis to select the most promising miRNA biomarker candidates for diabetic bone disease.

## 2. Results

### 2.1. Study Design

ZDF fa/fa rats spontaneously develop T2DM between the age of weeks 9 and 11 due to a homozygous mutation of the leptin receptor [10], whereas here, wild-type (WT) rats served as the non-diabetic control. Cryo-preserved serum and ulna bone tissue samples from 27 male ZDF rats (ZDF fa/fa) and 5 Zucker lean rats (ZDF +/+), known as WT rats, were obtained. The bone phenotype of all of the animals that were used for this study had previously been investigated [11,12,13] (Table 1). A 3 mm cross-sectional subcritical defect was created at the femur midshaft in all of the rats, as described by Hamann et al. [11]. Following the defect, the WT animals received placebo (water), PTH, and anti-sclerostin treatments for 12 weeks, while ZDF rats received placebo, PTH, anti-sclerostin, and insulin treatment (Table 1). Serum was collected from all the rats, while ulna bone samples were collected from both ZDF placebo groups and the PTH group.

### 2.2. Animal Phenotype Analysis

The animal phenotypes were re-analyzed from the previous data published by Hamann and Picke et al. [11,12,13]. Body weight (Figure 1A) did not significantly change in any group of ZDF rats, while glucose levels (Figure 1B) were significantly increased in all of the ZDF groups and was not rescued by any treatment. CTX (C-terminal telopeptide) levels showed a clear tendency to be increased in placebo-treated ZDF rats compared to WT rats (Figure 1C). The analysis of femoral bones (Figure 1D–F) was performed in all of the groups, but we lack data for anti-sclerostin-treated ZDF rats in this study. Only insulin treatment, not PTH treatment, increased the defect-healing capacity of the left femur (Figure 1D). The total and trabecular bone mineral density (BMD) (Figure 1E,F) showed significant increases under the PTH treatment, but not under insulin treatment. Analyses of L4 vertebra (Figure 1G,H) included anti-scl-treated ZDF rats and showed no effect of insulin treatment but a clear effect of PTH and an even stronger effect of anti-scl treatment on the total and trabecular BMD.

### 2.3. MiRNA Dysregulation in Serum and Bone Tissue of ZDF Rats

NGS analysis was performed on all 32 serum samples and 16 bone tissue samples obtained from WT and ZDF rats under the different treatments (Table 1). The NGS data quality was checked on the basis of total reads obtained per sample and relative read mapping against miRNA reads (Appendix A). The total reads > 17 nt (after quality filtering and adapter trimming) ranged between 2 and 10 million reads, with 5% reads on average mapping against miRNAs. Overall, 629 distinct miRNAs were detected in serum and bone tissue, of which, more than 250 miRNAs were detected per sample. Of these, more than 100 miRNAs showed a read count of >10 reads (Appendix A). First, we analyzed miRNA changes in serum and bone in five animals in each of the placebo-treated ZDF and WT rat groups using an adjusted *p*-value cut-off of 0.2 (FDR < 20%) to identify differentially expressed miRNAs. Out of the 629 detected miRNAs, 161 serum and 208 ulna bone miRNAs were included in the analysis after filtering for low-abundance miRNAs (Figure 2). In the serum of placebo-treated ZDF rats, we identified eight significantly upregulated and four downregulated miRNAs (Figure 2A) compared to WT rats.

In bone tissue, the same group comparison identified 10 significantly upregulated and 27 downregulated miRNAs (Figure 2B). Among all of the miRNAs dysregulated in the serum and bone tissue of ZDF rats, only one miRNA (rno-miR-199a-3p) was found to be shared (downregulated) in both compartments (Figure 2C). In serum, a log_2_ fold change (log_2_FC) of −1.8 (FDR = 0.028), and in ulna bone, a log_2_FC of −1.2 (FDR = 0.16) was observed.

### 2.4. Effect of PTH, Anti-Sclerostin, and Insulin Treatment on MiRNA Serum Levels and Bone Tissue Expression

To compare serum miRNA patterns in placebo vs. treated ZDF rats, the unsupervised analysis of all serum samples was performed based on the expression values of miRNAs with changes in placebo-treated ZDF vs. WT rats (top 30 with *p*-value < 0.1). The heatmap and hierarchical clustering (Appendix A) shows two main clusters: one consisting of WT animals as well as insulin- and anti-sclerostin-treated ZDF animals, and a second cluster consisting of placebo- and PTH-treated ZDF samples, indicating that insulin and anti-sclerostin treatment but not PTH treatment could reverse miRNA changes in serum.

Indeed, a groupwise comparison showed that anabolic treatment using anti-sclerostin resulted in the significant (FDR < 0.2) regulation of seven miRNAs (four up, three down) compared to placebo-treated ZDF rats (Figure 3A). The treatment of ZDF animals with insulin significantly (FDR < 0.2) changed the levels of 42 miRNAs (18 up, 24 down) compared to placebo-treated ZDF rats (Figure 3B). To identify which miRNAs dysregulated in placebo-treated ZDF rats compared to WT rats could potentially be rescued by treatment to normal WT levels, we overlapped those dysregulated miRNAs in ZDF rats with the miRNAs dysregulated by treatment in ZDF rats compared to placebo-treated ZDF rats. By doing so, we identified one miRNA (rno-miR-145-5p) that overlapped with the effects of anti-sclerostin treatment, which could potentially rescue this miRNA to WT levels, as shown in the upper central area of the VENN diagram (Figure 3C). Anti-sclerostin treatment also induced the downregulation of five miRNAs in WT rats compared to the placebo treatment, including miR-199a-3p and miR-145-3p (Appendix A). In the bottom and left area of the VENN diagram, seven miRNAs (rno-miR-802-5p, rno-miR-122-3p, rno-miR-375-3p, rno-miR-27a-5p, rno-miR-31a-5p, rno-miR-192-5p, and rno-miR-122-5p) were found to overlap between ZDF vs. WT and ZDF insulin vs. ZDF placebo, suggesting a potential rescue effect of insulin treatment in miRNAs to normal wild-type miRNA levels (Figure 3C). Unlike anti-sclerostin and insulin treatments, anabolic treatment by PTH did not induce any significant dysregulation in circulating miRNAs in ZDF or in WT rats (Appendix A). In ulna bone tissue, PTH treatment did not induce any significant changes in the miRNA patterns as compared to the placebo treatment (Appendix A), but it did show profound effects in WT rats compared to the placebo treatment (Appendix A). In total, 19 miRNAs (9 up, 10 down) were differentially regulated and, similar to insulin treatment in ZDF rats, a downregulation of miR-31a-3p was observed.

### 2.5. Anti-Scl and Insulin Treatment Rescue Serum Levels of Different MiRNAs in ZDF Rats

Out of the 12 dysregulated miRNAs in placebo-treated ZDF rats compared to WT rats, 4 (rno-miR-199a-3p, rno-miR-218a-5p, rno-miR-320-3p, and rno-miR-100-5p) were not changed under anti-Scl or insulin treatment (Figure 4A–D). Rno-miR-145-5p was upregulated in placebo-treated ZDF rats (vs. WT) and downregulated by anti-Scl treatment (Figure 4E and Table 2). Finally, insulin treatment in ZDF rats reversed the circulating levels of seven miRNAs to the levels observed in placebo-treated WT rats (Figure 4F–L and Table 2).

In order to investigate the cellular origin and putative biological function of miRNAs found to be dysregulated in serum as well as in bone tissue, we analyzed miRNA tissue enrichments and constructed mRNA target interaction networks.

### 2.6. Gene Target and Network Analysis Comprises Pathways Involved in Bone Biology and Osteoporosis Potentially Targeted by the Dysregulated MiRNAs in ZDF Rats in Serum and in Ulna Bone

In order to obtain a miRNA-targeted gene network, we mapped 12 circulating miRNAs that were differentially dysregulated in ZDF rats compared to WT rats (Figure 2A) to their experimentally confirmed target mRNAs using miRNet 2.0. The resulting list of gene–miRNA interactions was filtered for genes with at least four gene–miRNA interactions, narrowing down this list to 183 targeted genes. Next, pathway enrichment analysis was performed on this list of genes based on the KEGG, Reactome, and DisGeNET classifications to explore known biological pathways and diseases in which the targeted genes are involved. From the list of enriched (FDR < 0.05) pathways, we selected those relevant to bone biology and bone disease (“pathways of interest”, Appendix A). We identified 34 target genes in the pathways of interest of which 15 genes were present in more than three pathways. Prominent genes were *RHOA*, *AKT1*, *RAC1*, *MDM2*, *CDKN1A*, and *VEGFA*, which play important roles in TGF-beta and mTOR signaling (KEGG database), signaling by Wnt and by NOTCH (Reactome database), and in complications in diabetes mellitus (DisGeNET database).

Using the same approach, we mapped 37 miRNAs differentially dysregulated in the ulna bones of placebo-treated ZDF rats vs. placebo-treated WT rats (Figure 2B) against experimentally verified targets. By using a degree filter of two miRNA interactions for each gene, 161 targeted genes were identified, of which, 38 genes were associated with pathways of interest obtained through KEGG, Reactome, and DisGeNET databases. Sixteen genes were frequently present (more than three times) in the pathways of interest (Table 3), such as *SMAD2*, *CRK*, and *YWHAB*, which play important roles in TGF-beta, neurotrophin, and MAPK signaling (KEGG database), and also NGF signaling via TRKA and VEGFA-VEGFR2. Other genes of interest were *CDKN1B*, *AGO1*, and *MDM2*, which are involved in cellular senescence, as well as DAP12 and EGFR signaling pathways (Reactome database). *NOTCH2* and *CCND1* also appeared in our analysis in the Reactome database, being involved in NOTCH signaling and, interestingly, in the DisGeNET database, being involved in bone complications such as acro-osteolysis and pathological fractures. To elucidate which were the potential genes jointly targeted by both circulating and bone miRNAs, we overlapped the frequently present genes (present more than three times) in the pathways targeted by dysregulated miRNAs in ZDF rats in both serum and ulna bone (Appendix A), which showed three overlapping genes: *MDM2*, *TNRC6A* (Trinucleotide Repeat Containing 6A), and *TNRC6B* (Trinucleotide Repeat Containing 6B).

*MDM2*, *TNRC6A*, and *TNRC6B* are involved in pathways relevant to bone physiology and are targeted by dysregulated circulating miRNAs such as cellular senescence, SCF-KIT signaling, DAP12 signaling, and particularly *TNRC6A* and *TNRC6B*, which are also involved in NOTCH and Wnt signaling pathways (Appendix A). On the other hand, our network analysis showed that *MDM2*, *TNRC6A*, and *TNRC6B* are targeted by dysregulated bone miRNAs while being involved in cellular senescence, PI3K/AKT activation, and NOTCH, TGF-beta, NGF, SCF-KIT, DAP12, and EGFR signaling pathways (Table 3).

### 2.7. Gene Target and Network Analyses Reveal Several Pathways Involved in Bone Biology and Osteoporosis as Potential Targets for the Dysregulated MiRNAs in ZDF Rescued by Insulin

In order to identify the putative underlying mechanisms of the seven miRNAs rescued by insulin in T2D-based bone pathology, we performed a gene network analysis of genes targeted by these circulating miRNAs (Table 2). Again, experimentally verified targets were identified through miRNet 2.0, and the list of gene–miRNA interactions was filtered for genes with at least two miRNA interactions, resulting in 282 target genes.

After gene set enrichment analysis against KEGG, Reactome, and DisGeNET annotations, a total of 54 genes were selected for their presence in pathways of interest (Figure 5), of which, 19 genes were frequently (more than three times) present in the identified pathways (Table 4). For example, *RHOA*, *AKT1*, and *MYC* were identified, which play important roles in pathways from both KEGG and Reactome databases such as Wnt, VEGFA, NGF, and SCF-KIT signaling (Table 4). Other genes such as *XIAP*, *ACTB*, *RICTOR*, and *AGO2* and *AGO3* are involved in pathways such as apoptosis and focal adhesion (KEGG database) and NOTCH signaling, FGFR signaling, and Ca2+ pathways (Reactome database). The *LRP5* gene is involved in Wnt signaling (KEGG and Reactome databases) and also in several bone complications such as osteopetrosis, increased fracture rate, and osteoporosis with pseudoglioma (DisGeNET database).

### 2.8. Cell-Type Enrichment Analysis Reveals Distinct Putative Donor Cell Types of the Dysregulated and Rescued MiRNAs in ZDF Rats

In principle, circulating miRNAs can originate from any cell type. However, approximately 5–10% of all miRNAs are transcribed at higher levels in only a few cell types, resulting in cell-type-specific miRNAs (“cell-type enrichment”). In order to determine a putative cellular origin of miRNAs dysregulated in serum, we retrieved the cell expression profiles as well as the associated cell ontology terms for all significantly dysregulated circulating miRNAs identified in this study from the FANTOM5 database (Appendix A).

First, we analyzed the potential origin of circulating miRNAs that changed in ZDF rats versus WT rats but were not rescued by any treatment. Surprisingly, miR-100-5p and miR-199a-3p were found to be highly enriched in mesenchymal stem cell (MSC)-derived tissues, showing the highest expression in osteoblasts, fibroblasts, chondrocytes, adipocytes, and muscle cells (Appendix A). Similarly, miR-218-5p was found to be highly expressed in MSC derived tissues such as stroma, muscle, and fat, but not bone (Appendix A). Last, miR-320a was shown to be enriched in epithelial cells, but also in MSCs and their derived cells thereof such as chondrocytes and smooth muscle cells (Appendix A). This miRNA was also transcribed at lower levels in skeletal muscle cells and osteoblasts.

Next, we focused on the seven circulating miRNAs that were rescued by insulin treatment. From these, miR-375 (Figure 6A) was found to be enriched and highly expressed in the pituitary gland, while miR-192-5p (Figure 6B) appeared to be highly expressed in intestinal and prostate epithelial cells and in hepatocytes. miR-122-5p, as well as miR-122-3p, was found to be enriched in hepatocytes (Figure 6C,D). miR-31-5p appeared to be enriched in the epithelial cells of internal surfaces (Figure 6E) and was also transcribed at lower levels in MSCs and their derived cells. miR-145-5p (Figure 6F), the only circulating miRNA rescued by anti-sclerostin, was found to be highly enriched in MSCs and their derived cells, especially osteoblasts, fibroblasts adipocytes, and muscle cells.

## 3. Discussion

We hypothesized that the dysregulation of miRNA expression in bone tissue, as well as systemic dysregulation in serum, could be relevant to the phenotypes observed in the ZDF rat model of diabetic osteoporosis. Therefore, we used small RNA-sequencing analysis for the untargeted characterization of miRNA levels in serum and bone samples from WT as well as ZDF rats under anti-sclerostin, PTH, and insulin treatment.

### 3.1. MiR-199a-3p Is Jointly Downregulated in Serum and Bone Tissue of ZDF vs. WT Rats

Although several miRNAs were found to be dysregulated in the bone or serum of ZDF rats (Figure 2C), only one miRNA (miR-199a-3p) was found to be downregulated in both compartments. Cell-type enrichment analysis suggested that miR-199a-3p is mainly transcribed in mesenchymal-derived cells, including osteoblasts (Appendix A). This suggests that the systemic diabetic disease present in ZDF rats induces the downregulation of miR-199a-3p in bone tissue and other mesenchymal-cell-derived tissues, causing a lower release into the circulation from these tissues. High levels of miR-199a-3p were also reported in exosomes from human bone marrow MSCs [14], and the authors suggested that the transport of miR-199a-3p in MSC exosomes is a component of intercellular communication in the bone niche. When the decrease in bone mass and the increase in body weight shown by ZDF rats (Figure 1) is taken into consideration, this downregulation of miR-199a-3p in ZDF rats is in contrast with reports showing that the enhanced expression of miR-199a-3p promoted adipogenesis from bone-marrow-derived MSCs [15], while its downregulation enhanced the osteogenic differentiation of bone marrow MSCs in ovariectomized rats [16]. However, supporting our data, miR-199a-3p was upregulated in osteocytic areas in ovariectomized (OVX) mice and was also an inductor of catabolic bone cell autophagy in osteocyte-like MLO-Y4 cells [17].

### 3.2. Gene Network Analysis of Dysregulated MiRNAs in Serum or Bone of ZDF vs. WT Rats

Our gene target network analyses of the dysregulated miRNAs in the serum and in the ulna bone of ZDF versus WT rats revealed targeted genes involved in bone-related pathways (Table 3, Appendix A). Interestingly, these analyses also revealed that the genes implicated in bone complications were targeted by miRNAs dysregulated in bone (Table 3), while genes associated with complications in diabetes mellitus were targeted by miRNAs dysregulated in the circulation (Appendix A). This finding highlights a potential application of these miRNAs as biomarkers for diabetic bone disease.

We identified *MDM2*, *TNRC6A*, and *TNRC6B* to be jointly targeted by both circulating and bone miRNAs dysregulated by T2D in ZDF rats (Appendix A). Our network analysis with dysregulated miRNAs in the circulation (Appendix A) or in bone (Table 3) show these genes are involved in pathways relevant to bone physiology. We hypothesize that a paracrine effect of circulating miRNAs could synergize and enhance the targeting of those three genes in bone in the context of diabetes.

A recent study showed, using mice haploinsufficient for *MDM2* in their adipose stores, that *MDM2* regulates systemic insulin sensitivity, with these mice experiencing impaired insulin sensitivity [18]. Regarding its potential role in bone, high levels of *MDM2* in mice osteoblast and in the MG63 human osteosarcoma cell lines induced an increase in bone mineralization and also corrected aged-related bone loss in mice [19]. Recent findings identified that a *TNRC6A* gene variant in the Mizo population is involved in Mizo susceptibility to T2D [20]. Until now, there have been no reports showing any potential role of this gene on bone physiology. Another recent study identified a single-nucleotide polymorphism (SNP) in the *TNRC6B* gene being positively associated with lean mass [21]. Furthermore, this *TNRC6B* SNP was associated with spine BMD and increased risk of fractures, suggesting not only a potential role of the *TNRC6B* gene in lean mass development but also in bone physiology. Interestingly, the two aforementioned genes with a potential role in bones, *MDM2* and *TNRC6B*, were also among the most frequently present genes in Reactome database pathways targeted by miRNAs rescued by insulin treatment (Table 4).

### 3.3. Anti-Sclerostin Treatment

Neither anti-sclerostin nor PTH treatments affected body weight, serum glucose levels, or the metabolic function of WT and ZDF rats. This clearly indicates an insulin-independent but bone-specific mode of action for the observed bone quality improvement by these treatments [11,12]. Anti-sclerostin treatment fully reversed the negative effects of T2DM on bone mass and increased bone mass in ZDF rats compared to placebo-treated WT rats (Figure 1). We hypothesize that the underlying mechanism might be connected to the dysregulation of circulating miRNAs by anti-sclerostin such as miR-29b-3p (Figure 3A) and miR-145-5p, which are restored by anti-Scl treatment to the levels observed in WT rats (Figure 4E). MiR-145-5p could primarily originate from MSC lineages such as osteoblasts (Figure 6F).

We hypothesize that the decrease in circulating miR-145-5p under anti-sclerostin treatment is due to a potential downregulation of miR-145-5p in bone tissue. Anti-sclerostin treatment also significantly improved bone quality in WT rats compared to untreated WT rats [11] and induced the downregulation of five miRNAs (Appendix A), similar to its effect on circulatory miRNAs in ZDF rats. Interestingly, miR-145-3p was the most downregulated miRNA in WT rats under anti-sclerostin treatment when compared to untreated WT rats (Appendix A), similar to miR-145-5p in ZDF rats. Additionally, miR-145-3p is mainly expressed MSCs and their derived cells. Taken together, we hypothesize that the inhibition of sclerostin could have a direct impact on bone through the downregulation of miR-145-5p/3p transcription in bone tissue, resulting in reduced osteoclastogenesis by the upregulation of its target osteoprotegerin [22] and an increase in osteogenic differentiation by increasing Sp7 [23] and semaphoring 3A target levels [24].

### 3.4. PTH Treatment

PTH treatment reversed the adverse skeletal effects of T2DM in ZDF rats on total and trabecular bone mass (Figure 1). Compared to anti-sclerostin treatment, PTH treatment did not change bone turnover markers, was less potent at both cortical and trabecular sites of ZDF rat bones [12], and did not have any significant effects on circulating miRNAs in WT and ZDF rats (Appendix A). These lower or even non-significant effects of PTH compared with anti-sclerostin treatment may be related, as previously discussed by Hamann et al., to the duration of the PTH treatment, the used dose, or both [12]. In a previous study, ovariectomized (OVX) rats [25] were treated with a lower PTH dose for 12 weeks, and several miRNAs were identified to have been significantly regulated in femoral head tissue. Another potential explanation for this differential effect of the two anabolic treatments on bone phenotype and circulating miRNAs is the different molecular mechanisms of action of these treatments, since anti-Scl treatment is known to be not only pro-anabolic but also anti-catabolic for bone tissue, while PTH treatment is both pro-anabolic and pro-catabolic, with its anabolic action being stronger than the catabolic action when given intermittently as in this study [26].

Furthermore, the PTH treatment effects on bone were blunted in ZDF rats compared with WT rats [12], but no blunted effects on skeletal response in ZDF rats under anti-sclerostin treatment were observed [11]. This relative PTH resistance of ZDF rats could be, as previously proposed by Hamann et al., due to T2DM chronic inflammation or the modulation of PTH secretion and blunting of PTH effects by high-glucose levels observed in vitro in primary parathyroid cells and in osteoblasts, respectively [12]. A recent study also showed similar PTH resistance and blunting of the PTH anabolic effects on bone due to suppressed Wnt signaling in a streptozotocin-induced T1D mouse model [27]. This blunting effect of diabetes on PTH is also reflected by our data showing miRNA dysregulation induced by PTH treatment in the ulna bones of WT rats but not in the bones of ZDF rats (Appendix A), which also supports the concept that the partial mediation of the PTH anabolic effects on the bones of WT rats could be through the regulation of bone miRNAs.

Interestingly, PTH treatment induced an upregulation of miR-100-5p and a downregulation of miR-31a-5p in WT rat bones, which were downregulated and upregulated, respectively, in the serum of ZDF rats compared to WT rats (Appendix A). Of note, miRNA abundance in the ulna bones of WT rats was not reflected in the serum of WT rats when compared to PTH-treated animals (Appendix A). A similar phenomenon was observed in the miRNA dysregulation of untreated ZDF rats compared to WT rats, being higher in ulna bones than in serum, with only one miRNA (miR-199-3p) being dysregulated in both compartments (Figure 2C). These results indicate that miRNA dysregulation in bone is generally not well reflected in circulation.

### 3.5. Insulin Treatment

Measurements of glycosylated hemoglobin shown by Hamann et al. indicated that, even if glucose levels do not change with insulin (Figure 1B), insulin treatment elicits metabolic recovery in ZDF rats [13]. As indicated by Picke et al., bone defect regeneration may be affected by typical T2DM molecular imbalances [13], which are reverted by insulin treatment but not by the anabolic treatments anti-sclerostin and PTH. Because of this, anti-Scl and PTH treatment partially increased defect filling [11,12], whereas insulin treatment showed a higher effect by increasing defect regeneration up to the control level [13]. Insulin treatment also showed the strongest impact on circulating miRNAs in ZDF rats (Figure 3B) despite their unaltered glucose levels at the time of sample collection (Figure 1B), indicating that the effect of insulin on miRNA regulation is more persistent than on glucose.

Gene target network analysis indicates that the seven circulating miRNAs rescued by insulin could play a potential role in bone disease by targeting genes involved in pathways related to bone physiology, such as *LRP5*, which is also involved in certain bone diseases and complications (Figure 5, Table 4). Interestingly, *LRP5* was found to be associated not only with bone physiology but also with metabolic alterations, showing in a mouse model of diabetes that a gain-of-function *Lrp5* mutation improved bone quality and delayed hyperglycemia [28].

Cell-type enrichment analysis of the circulating miRNAs rescued by insulin in ZDF rats (Figure 6) revealed that these miRNAs are not specifically expressed in bone tissue but hepatocytes and gastrointestinal tissues, fitting with the indirect effect of insulin on bone healing. However, the rescue of circulating miRNAs by insulin could have a paracrine action in bone tissue, improving the bone-healing capability of ZDF rats (Figure 1D). Of note, none of the miRNAs were found to be expressed by the pancreas even though they are rescued by insulin, because the FANTOM5 browser lacks data for pancreatic tissue.

Previously published data support this hypothesis: miR-27a promoted osteogenic differentiation in rats [29] and also by its transport to osteoblast cells in extracellular vesicles [30], while miR-802, a well-known marker of obesity with a role in T2DM [31,32], was also shown to promote osteosarcoma cell proliferation [33]. MiR-375 is a known marker of T2DM [32,34] and also predicts T2DM in patients when its levels are low years before the onset of the disease [35]. Furthermore, serum levels of miR-375 were, similar to our ZDF data, diminished in T2D patients treated with insulin and also increased in T2D patients with poor glycemic control compared to T2D patients with adequate glycemic control [36]. Another study showed that circulating miR-375 and miR-122-5p are potential biomarkers for bone mass recovery after parathyroidectomy in patients with primary hyperparathyroidism [37]. Furthermore, miR-375 was shown to inhibit osteogenesis in a C2C12 cell model of osteogenic differentiation [38]. Similarly, teriparatide alleviates osteoporosis by promoting the osteogenesis of hMSCs via the miR-375/RUNX2 axis [39]. However, miR-375 has also been found to be an inducer of osteogenesis [40] and a promoter of bone regeneration [41] in human adipose MSCs and human bone marrow MSCs, respectively. It is unclear how these contradictory data might be reconciled.

Circulating miR-122 is also a known biomarker for T2D [42,43] and is strongly associated with the risk of developing T2D in the general population [44]. Regarding its potential role in bones, the overexpression of miR-122-3p was shown to inhibit the osteogenic differentiation of mouse-adipose-derived stem cells [45]. Its reverse complement and more abundant strand, miR-122-5p, acts as an inhibitor of MSC differentiation in mouse [46] and of osteoblast proliferation and differentiation in osteoporotic ovariectomized rats [47]. Furthermore, circulating miR-122-5p was shown to be a potential marker of osteoporosis in patients, being associated with fragility fractures and with low BMD [48].

miR-192-5p has been reported to be a candidate biomarker of the prognosis of diabetes [49], but has not been found yet to have a specific role in bone tissue. Finally, miR-31-5p is, interestingly, downregulated by insulin in ZDF rat serum and also by PTH in the ulna bones of WT rats. MiR-31-5p has a well-known role in bone formation as an inhibitor of *Wnt* [50], *SATB2* [51], and *FRZD3* [52], and therefore of osteoblastic differentiation, while promoting osteoclastic differentiation [51]. This role seems very well conserved evolutionarily, as it is involved in exoskeleton formation in sea urchins [53]. Furthermore, miR-31 was upregulated in another study in patients with osteoporosis, and also during RANKL-induced osteoclastogenesis, while miR-31 inhibition in murine bone-marrow-derived macrophages impaired bone resorption [54]. Last, recent studies showed that miR-31-5p was downregulated in patients with osteoporosis due to a mutation in a Wnt pathway ligand [55] and was also associated with incident fragility fractures in older diabetic women, meaning it could possibly be used as a prognostic biomarker for fracture risk in the context of T2D [9]. To summarize the findings regarding these circulating miRNAs, their paracrine effect on bones through their rescue by insulin could synergize with the rescue of the metabolic and inflammatory phenotype of ZDF rats, enhancing their bone-healing capability.

### 3.6. MiRNAs Not Rescued by Any Treatment

Finally, we identified four circulating miRNAs that were dysregulated in ZDF rats but not rescued by any treatment: rno-miR-199a-3p, rno-miR-218a-5p, rno-miR-320-3p, and rno-miR-100-5p (Figure 4A–D). Together with miR-145-5p, miR-320a and, especially, miR-199a-3p and miR-100-5p, they were the only biomarker candidates in this study that are likely released from bone tissue, even though none of the treatments could rescue their levels in the serum of ZDF rats to normal WT levels, creating the possibility that a different mechanism induces their dysregulation in ZDF rats rather than their characteristic insulin resistance, hyperglycemia or impaired osteoblastic function. Unlike anti-sclerostin treatment with its rescue effect on miR-145-5p, and PTH treatment inducing miR-100-5p upregulation in WT rat bones (Appendix A), insulin treatment was not able to significantly alter any dysregulated circulating miRNAs in ZDF rats enriched in bones (Figure 3C). This could indicate again that, unlike the anabolic treatments, the effect of insulin on bone was not a direct effect and more based on a potentially paracrine effect of the insulin-induced rescue of circulating miRNAs in synergy with the metabolic regulation of diabetes.

### 3.7. Contribution to the Field and Limitations of This Study

To our knowledge, this is the first report of an effect of anti-sclerostin treatment on circulating miRNAs. With this study, the effects of different anti-osteoporotic treatments in the context of diabetes on circulating miRNAs were also described and compared for the first time. This study showed a potential link between miRNA dysregulation in circulation and in bone tissue and the improvement of bone quality in rats with diabetic osteoporosis. However, this study had several limitations. The first one is the lack of a high number of animals per group and of biochemical markers, histomorphometric markers, and miRNA detection at intermediate stages of the treatment. This limitation means it was not possible to have a high enough number of samples from the different rat groups with measurements of anabolic bone turnover markers. Furthermore, the different studies did not share the same analysis of these markers (P1NP for the insulin study, P1NP and osteocalcin for the anti-Scl study, and carboxylated osteocalcin for the PTH study). Last, even though the ZDF rat model shares a number of common metabolic disturbances (leptin signaling deficiency, insulin resistance, hyperglycemia, and hyperinsulinemia) with patient populations, it lacks the increased BMD and bone cortical porosity characteristic of patients with T2DM.

## 4. Materials and Methods

### 4.1. Animals

For this study, rats were purchased from Charles River Laboratories at 9 weeks of age and had received a high-fat, high-carbohydrate chow diet (Purina 5008) starting at 11 weeks of age for 12 weeks. All rats were male and had an initial weight range of 350–400 g. After 12 weeks, all rats were sacrificed under general anesthesia. All invasive procedures (such as the subcritical defect or the administration of the different treatments) were performed at the TUD (Technical University of Dresden, Germany) and approved by the local Institutional Animal Care Committee (approval number: 24D-9168.11-1/2008-30). Body weight measures, serum analysis, and the assessment of bone mass, bone microarchitecture, and bone-defect healing were performed at the TUD in all rats as previously described by Hamann and Picke et al. [11,12,13]. Briefly, glucose levels were measured using a Roche Modular PPE analyzer, while CTX levels were detected using an immunoassay kit (Immundiagnostik Systems, GER). BMD for the intact femur and the vertebral body was analyzed while blinded by microcomputed tomography (μCT) using a vivaCT40 (ScancoMedical, Wangen-Brüttisellen, Switzerland). Bone inside the defect zone of the left femur was measured ex vivo using a μCT (vivaCT40, ScancoMedical) to detect the BV/TV of newly formed bone. To investigate changes in circulating and bone miRNA expression as a consequence of the ZDF diabetic phenotype and in response to treatment, serum and ulna bone samples were harvested at week 12 after treatment at the Bone Lab located in the TUD and used for RNA extraction. RNA extraction from serum and ulna bone and NGS were performed at TAmiRNA GmbH for a genome-wide screen of miRNA levels in a subset of 32 serum samples (5× WT-Placebo, 5× ZDF-Placebo, 5× WT-PTH, 5× ZDF-PTH, 4× ZDF-Insulin, 4× WT-anti-Sclerostin, and 4× ZDF-anti-Sclerostin) and 16 ulna bone samples (4× WT-Placebo, 4× ZDF-Placebo, 4× WT-PTH, and 4× ZDF-PTH). Bone samples were collected from the same rats used for serum collection, with the exception of 2 out of 4 bone samples from PTH-treated ZDF rats.

### 4.2. RNA Extraction from Serum

Total RNA was extracted from 100 μL of serum using the miRNeasy Mini Kit (Qiagen, Hilden, Germany) as described by Kocijan et al. [25]. Then, 100 μL of each serum sample was mixed with 1000 μL of Qiazol and 1 μL of a mix of 3 synthetic spike-in controls (Exiqon, Copenhagen, Denmark). After a 10-min incubation at room temperature, 200 μL of chloroform was added to the lysates followed by centrifugation at 12,000× *g* for 15 min at 4 °C. Then, 650 μL of the upper aqueous phase was transferred to a miR-Neasy mini column, where RNA was precipitated with 750 μL of ethanol followed by automated washing with RPE and RWT buffer in a QiaCube liquid handling robot. Finally, total RNA was eluted in 30 μL of nuclease-free water and stored at −80 °C.

### 4.3. RNA Extraction from Ulna Bone

Ulna bones were harvested from all rats and homogenized at the Bone Lab in the TUD (Dresden, Germany), and samples were immediately snap frozen in liquid nitrogen and stored at −80 °C. For RNA extraction, tissue homogenization was performed with frozen bones by using a mortar and a pestle containing liquid nitrogen followed by the addition of 1000 μL of Trizol. Tubes containing the bone sample and 1000 μL of Trizol were put back in liquid nitrogen. At TAmiRNA GmbH, bone fragments were collected from the tubes and transferred to new tubes containing Qiazol. The bone fragments were then further lysed with a TissueRuptor (Qiagen, Hilden, Germany). After lysis, the tubes were centrifuged at 8600× *g* for 30 s, and the resulting supernatant was used for total RNA extraction using the miRNeasy Mini kit (Qiagen, Hilden, Germany) as per the manufacturer’s recommendation.

### 4.4. Library Preparation for Small RNA-Seq

Based on previous studies performed by Khamina et al. [56], library preparation was performed using RealSeq-Biofluids Plasma/Serum miRNA Library kit for Illumina sequencing (RealSeq Biosciences, 600-00048; protocol 20181220_RealSeq-BF_CL) according to the manufacturer’s protocol. Briefly, 8.5 μL of extracted RNA was used as input. Adapter-ligated libraries were circularized, reverse transcribed and amplified. Library PCR was performed using 18–21 cycles with Illumina primers included in the kit. In total, 32 miRNA libraries were prepared from serum samples, and 16 miRNA libraries were prepared from ulna bone samples. All 48 libraries were analyzed for library fragment distribution using the Agilent DNA 1000 kit (Agilent Technologies, 5067–1504, Waldbronn, Germany) with Agilent DNA1000 reagents (Agilent Technologies, 5067–1505, Waldbronn, Germany). The generated libraries were pooled in an equimolar proportion, and the obtained pool was size-selected with the BluePippin system using a 3% agarose cassette with a target range of 100–250 kb (Sage Science, BDQ3010, Berverly, MA, USA) to remove DNA fragments outside of the target range. The pooled and purified libraries were analyzed for fragment distribution on an Agilent High Sensitivity DNA kit (Agilent Technologies, 5067–4626) with Agilent High Sensitivity DNA reagents (Agilent Technologies, 5067–4627). The library pool was then sequenced on an Illumina NextSeq550 (single-read, 75 bp) according to the manufacturer’s protocol at the Vienna BioCenter Core Facilities (VBCF), Vienna, Austria.

### 4.5. MiRNA Target Network Construction

The 12 significantly dysregulated circulating miRNAs and the 37 miRNAs significantly dysregulated in bone tissue in ZDF rats, as well as the 7 circulating miRNAs rescued by insulin, were used to construct a target network using the online tool miRnet 2.0 (https://www.mirnet.ca/miRNet/upload/MirUploadView.xhtml; accessed on 15 January 2022). Genes listed in miRTarBase v8.0, TarBase v8.0, and miRecords were selected for network construction. The degree filter was set to 1/3 of the analyzed miRNAs; hence, only gene target nodes with at least 4 connections for the 12 dysregulated circulating miRNAs, 12 connections for the 37 miRNAs dysregulated in bone, and 2 connections for the 7 circulating miRNAs rescued by insulin remained in the network. The KEGG, Reactome, and DisGeNET databases were used for pathway enrichment, using all genes identified in the network and hypergeometrical testing. For each of the databases used, pathways of interest (with an FDR < 20%) were selected based on their known role in bone biology or relation with bone quality and osteoporosis. For a simpler visualization of the network, all gene target nodes that were not present in the selected pathways of interest were hidden in the network. For the network analysis made with the dysregulated miRNAs in both serum and ulna bone, genes that appeared in at least three of the selected pathways of interest were considered to be frequently present (and most relevant) in our analysis and were overlapped in VENN diagrams (Appendix A).

### 4.6. Cell-Type Enrichment Analysis

FANTOM (Functional Annotation of the Mouse/Mammalian Genome) is an integrated expression atlas of miRNAs. It uses sRNA sequencing data across a wide variety of human and mouse samples to create an miRNA expression atlas for humans and mice and annotates each miRNA based on its expression profile across cell types [57]. Using the FANTOM5 browser (https://fantom.gsc.riken.jp/5/suppl/De_Rie_et_al_2017/vis_viewer/#/about; accessed on 20 January 2022), we could find and visualize the expression profile of human mature miRNAs across tissues and cell types as measured in FANTOM5 and the cell ontology terms associated with each mature miRNA.

### 4.7. Statistical Analysis

Statistical differences in bone and metabolic phenotypes between all groups of rats were determined using GraphPad Prims v9.2.0. The statistical significance of group differences was assessed using 1-way-ANOVA in conjunction with Holm–Sidak tests for multiple comparisons. *p*-values were only indicated for the comparisons between WT and ZDF rats within each treatment group, and between both placebo-treated WT and ZDF rats and ZDF rats under the rest of treatments.

The analysis of RNA-Seq data was performed with the software package MiND, a data analysis pipeline that generates overall QC data, unsupervised clustering analysis, normalized miRNA count matrices, and differential expression analysis based on raw NGS data [58]. The overall quality of the next-generation sequencing data was evaluated automatically and manually with fastQC v0.11.9 and multiQC v1.10. Reads from all passing samples were adapter trimmed and quality filtered using cutadapt v3.3 and filtered for a minimum length of 17nt. Mapping steps were performed with bowtie v1.3.0 and miRDeep2 v2.0.1.2, whereas reads were first mapped against the genomic reference Rnor.6.0 provided by Ensembl, allowing for two mismatches, and subsequently, miRBase v22.1, filtered for miRNAs of rno only, allowing for one mismatch.

For a general RNA composition overview, non-miRNA mapped reads were mapped against RNAcentral and then assigned to various RNA species of interest. The statistical analysis of preprocessed NGS data was conducted with R v4.0 and the packages espheatmap v1.0.12, pcaMethods v1.82, and genefilter v1.72. The differential expression analysis with edgeR v3.32 used the quasi-likelihood negative binomial generalized log-linear model functions provided by the package. The independent filtering method of DESeq2 was adapted for use with edgeR to remove low-abundance miRNAs and thus optimize the false discovery rate (FDR) correction. Scatterplots present the distribution of values as well as mean ± standard deviation. Hierarchical clustering with the associated heatmap was performed with the ClustVis online tool.

## 5. Conclusions

While metabolic treatment with insulin mainly rescued circulating miRNAs enriched in metabolic organs (such as the liver or pituitary gland), bone anabolic treatments mainly regulated miRNAs enriched in bone and its connected tissues (such as muscle or fat tissue).

Anti-sclerostin treatment could downregulate miR-145-5p/3p transcription in bone tissue, being reflected in the circulation. This would result in reduced osteoclastogenesis and an increase in osteogenic differentiation, contributing to the enhancing effects that anti-sclerostin treatment has on bone mass.

The effects of PTH treatment in the bone phenotype of WT rats may be indicated by and dependent on the regulation of bone miRNA levels, which is not reflected in the circulation, similar to the phenomena observed in ZDF rats when compared to WT rats.

We cannot conclude that the bone enhancement of ZDF rats is reflected or dependent on PTH-induced miRNA regulation, probably due to:-The previously described PTH resistance induced by diabetes in the ZDF rat model, with a blunting effect of T2DM over PTH.-The PTH dose probably being too high or too extended in time. This is also shown when compared to other studies that used ovariectomized (OVX) rats, resulting in a shorter time effect in miRNA patterns in ZDF rats compared to OVX rats.

Unlike the anabolic treatments, the effect of insulin on bone was not a direct effect and was more based on the potentially paracrine effect of the insulin-induced rescue of circulating miRNAs in synergy with the metabolic regulation of diabetes.

## Figures and Tables

**Figure 1 ijms-23-06534-f001:**
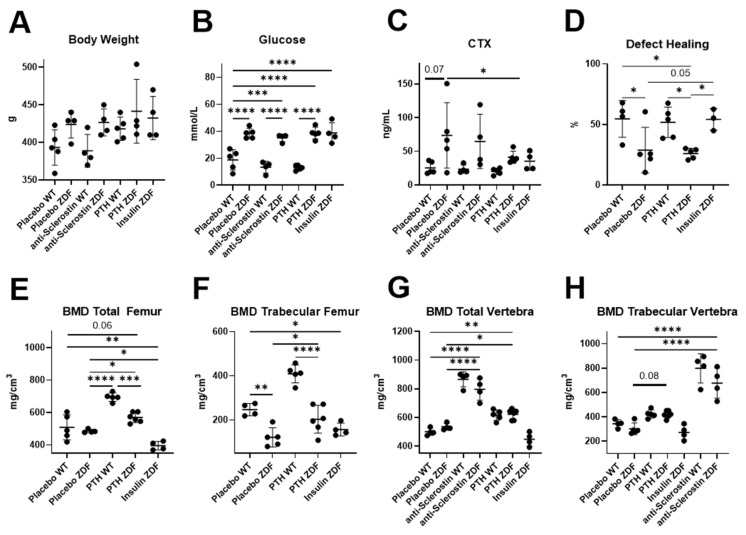
Re-analyzed characterization of animal phenotypes. Metabolic parameters (**A**–**C**), as well as femoral (**D**–**F**) and vertebral bone parameters (**G**,**H**), were evaluated. Ex vivo micro-CT analyses were performed to determine BV/TV and BMD (placebo WT, *n* = 5; placebo ZDF, *n* = 5, PTH ZDF, *n* = 5; insulin ZDF, *n* = 4; anti-scl ZDF, *n* = 4). Scatterplots show mean ± SD. Two-way ANOVA analysis was performed. *p* < 0.1 shown in numeric values, * *p* < 0.05, ** *p* < 0.01, *** *p* < 0.001, **** *p* < 0.0001.

**Figure 2 ijms-23-06534-f002:**
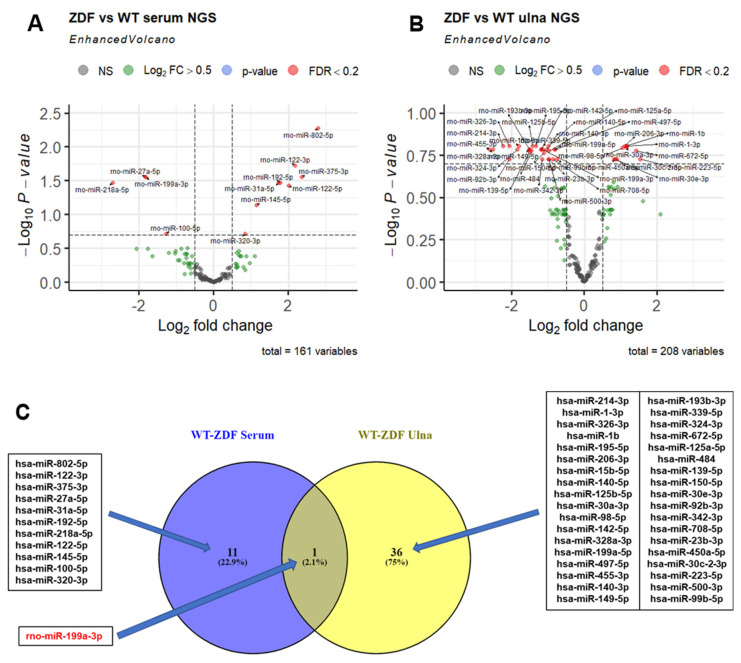
NGS-based discovery of miRNA changes in serum and ulna bone tissue of ZDF rats compared to WT rats under placebo treatment. (**A**,**B**) Volcano plots depict the log2-transformed fold change and log10-transformed adjusted *p*-values for the contrast (WT placebo, *n* = 5; ZDF placebo, *n* = 5) in both serum and ulna bone compartments. miRNA effects with Benjamini–Hochberg-adjusted *p*-values < 0.2 (FDR < 20%) are highlighted as separate group in red. (**C**) Comparison of miRNAs dysregulated in serum and in ulna bone (FDR threshold of 0.2). VENN diagram with the overlapping miRNA between the two compared lists corresponding to the miRNAs dysregulated in the ZDF model in serum (left circle) and ulna bone tissue (right circle). miR-199a-3p that is dysregulated in the ZDF model in both compartments is highlighted in red.

**Figure 3 ijms-23-06534-f003:**
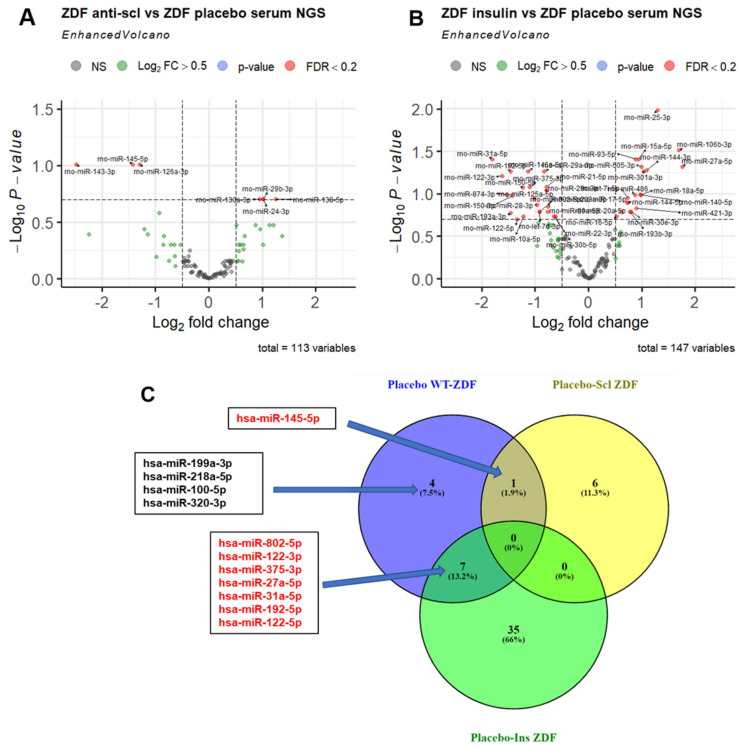
NGS-based discovery of miRNA changes in serum of ZDF rats under anti-sclerostin and insulin treatment compared to placebo treatment. (**A**,**B**) Volcano plots depict the log2-transformed fold change and log10-transformed adjusted *p*-values for the contrast (ZDF placebo, *n* = 5 vs. ZDF treatment, *n* = 4) in serum. miRNAs with Benjamini–Hochberg-adjusted *p*-values < 0.2 (FDR < 20%) are highlighted as separate group in red. (**C**) VENN comparison of miRNAs dysregulated in placebo-treated ZDF vs. WT rats (left circle) and miRNAs affected by either insulin (bottom circle) or anti-sclerostin treatment (right circle) (FDR < 20%). miRNAs that are dysregulated in the ZDF model and rescued by treatment are highlighted in red.

**Figure 4 ijms-23-06534-f004:**
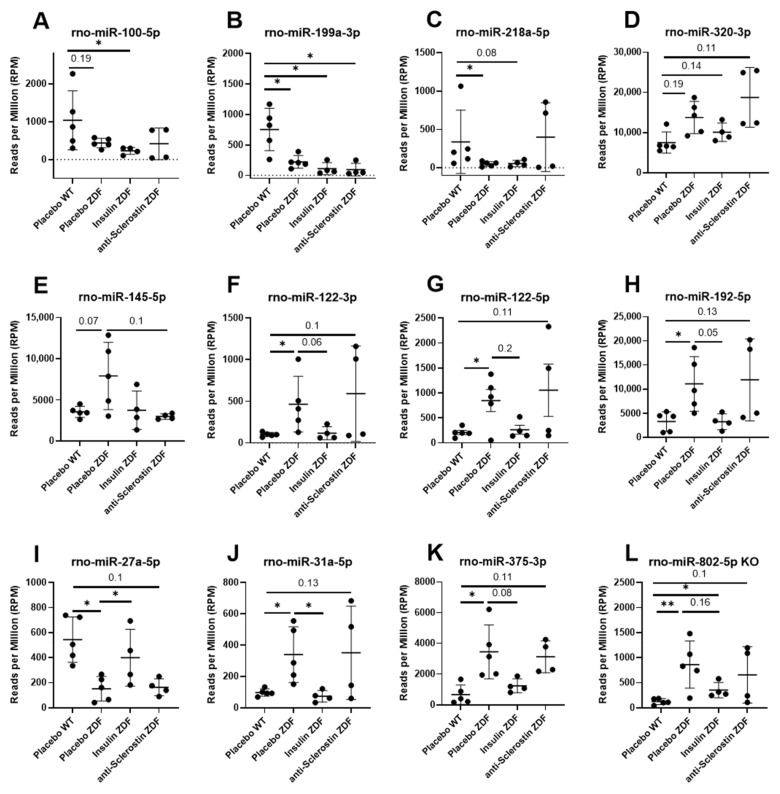
MicroRNA scatter plots. Representation of circulating miRNAs not rescued by any treatment (**A**–**D**), rescued by anti-sclerostin (**E**), or insulin (**F**–**L**) treatments. Placebo WT, *n* = 5; placebo ZDF, *n* = 5, PTH ZDF, *n* = 5; insulin ZDF, *n* = 4; anti-scl ZDF, *n* = 4. Scatterplots show mean ± SD. Testing was performed using edgeR as described in the Materials and Methods Section. FDR < 0.1 shown in numeric values, * FDR < 0.05, ** FDR < 0.01.

**Figure 5 ijms-23-06534-f005:**
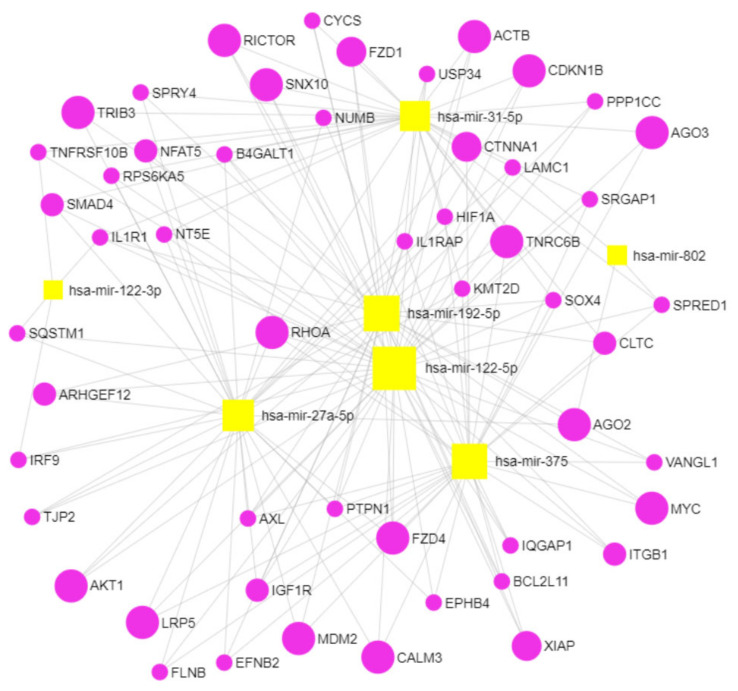
Target analysis of miRNAs rescued by insulin (Table 2) using miRnet. Seven miRNAs identified as significantly dysregulated by the diabetic phenotype in ZDF rats and rescued by insulin to WT levels were used to construct a target network using the online tool miRnet with a degree filter set to 2.

**Figure 6 ijms-23-06534-f006:**
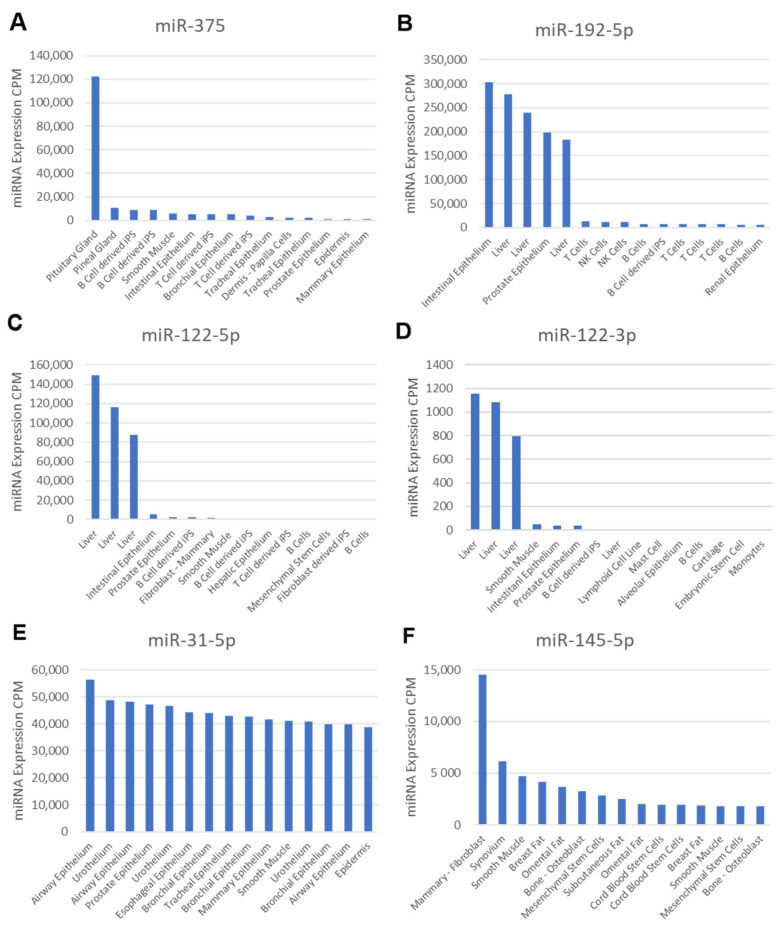
Tissue/cell-type specificity based on the FANTOM5 atlas browser of miRNAs rescued by treatment. Using the FANTOM5 browser, the expression profiles of dysregulated circulating miRNAs in ZDF rats rescued by insulin (**A**–**E**) or anti-Scl (**F**) across the top 15 tissues were obtained (no data were available for miR-802-5p). Y-axis indicates the level of expression in CPM of the miRNA for each analyzed cell/tissue sample; x-axis shows the position of each cell/tissue sample where the miRNA was detected, and the full name of which (given by FANTOM5) was summarized.

**Table 1 ijms-23-06534-t001:** Summary of the animals used for the study and the experimental design of the ZDF study [11,12,13]. In total, 18 ZDF rats and 5 wild-type rats, at the age of 11 weeks, underwent femur subcritical defect surgery. After 24 h, ZDF rats were randomized into a placebo treatment with vehicle solution (placebo, *n* = 5), PTH treatment (PTH, *n* = 5), insulin treatment (*n* = 4), or anti-sclerostin treatment (anti-scl, *n* = 5) for 12 weeks. Micro-CT, serum, and tissue analyses were performed as indicated in the Materials and Methods Section.

Genotype	Type of Treatment	Age at Beginning of Study	Treatment Administration	Treatment Duration	No. of Animals	Bone Defect	Serum/Ulna Collection at 12 Weeks	Original Reference
WT	Placebo	11 weeks	s.c. (daily)	12 weeks	5	yes	yes/yes	[12,13]
ZDF	Placebo	11 weeks	s.c. (daily)	12 weeks	5	yes	yes/yes	[12,13]
WT	Anti-Sclerostin	11 weeks	s.c. (twice a week) 25 mg/kg	12 weeks	4	yes	yes/no	[11]
ZDF	Anti-Sclerostin	11 weeks	s.c. (twice a week) 25 mg/kg	12 weeks	4	yes	yes/no	[11]
WT	PTH	11 weeks	s.c. (5 times a week) 75 µg/kg	12 weeks	5	yes	yes/yes	[12]
ZDF	PTH	11 weeks	s.c. (5 times a week) 75 µg/kg	12 weeks	5	yes	yes/yes	[12]
ZDF	Insulin	11 weeks	s.c. (daily) 0.5 IU at week 1–13 IU at week 12	12 weeks	4	yes	yes/no	[13]

**Table 2 ijms-23-06534-t002:** Cross-sectional regulation of the rescued circulating miRNAs via anti-sclerostin and insulin **treatment.** Differential expression of miRNAs in diabetic ZDF rats and after 12 weeks of treatment with anti-sclerostin and with insulin at the study’s end (WT placebo, *n* = 5; ZDF placebo, *n* = 5; ZDF anti-Scl, *n* = 4; ZDF insulin, *n* = 4).

		Placebo ZDF vs. Placebo WT		Treatment ZDF vs. Placebo ZDF	
Treatment	MiRNA	Log2FC	FDR	Log2FC	FDR
Anti-Scl	rno-miR-145-5p	1.144	**0.073**	−1.424	**0.098**
Insulin	rno-miR-802-5p	2.789	**0.005**	−0.918	**0.162**
	rno-miR-27a-5p	−1.823	**0.028**	1.75	**0.048**
	rno-miR-375-3p	2.349	**0.028**	−1.1	**0.083**
	rno-miR-192-5p	1.780	**0.034**	−1.45	**0.054**
	rno-miR-122-5p	2.012	**0.038**	−1.34	**0.199**
	rno-miR-122-3p	2.2	**0.019**	−1.6	**0.062**
	rno-miR-31a-5p	1.7	**0.034**	−1.8	**0.039**

**Table 3 ijms-23-06534-t003:** Pathway enrichment using the KEGG, Reactome, and DisGeNET databases. Several pathways significantly enriched with an adjusted *p* value < 0.2 (FDR < 20%) were identified for the thirty-seven dysregulated bone miRNAs. The top targeted genes most commonly present in the identified pathways are highlighted in bold (16 genes).

Database	Pathway	Hits	*p*-Value	FDR
KEGG	Focal adhesion	**CRK**; THBS1; **CCND1**; CCND2; **CRKL**; IGF1R; VEGFA; **JUN**; FLNA; DIAPH1; **PAK2**; **ACTG1**	6.1 × 10^−6^	1.9 × 10^−4^
	TGF-beta signaling	EP300; THBS1; **SMAD3**; **SP1**; **SMAD2**; BMPR2	6.4 × 10^−4^	0.0076
	Neurotrophin signaling	**CRK**; YWHAG; **CRKL**; **JUN**; YWHAQ; IRS4; **YWHAB**	8.9 × 10^−4^	0.0076
	Adherens junction	IGF1R; **SMAD3**; **SMAD2**; **ACTG1**	0.012	0.059
	MAPK signaling	**CRK**; **CRKL**; **JUN**; FLNA; RAPGEF2; ELK4; HSPA1B; **PAK2**	0.019	0.080
Reactome	Cellular Senescence	CDK6; **CDKN1B**; E2F3; **JUN**; **TNRC6A**; **SP1**; **AGO1**; **TNRC6B**; CBX6; **MDM2**	9.6 × 10^−6^	1.1 × 10^−4^
	Signaling by NOTCH	EP300; **CCND1**; E2F3; **TNRC6A**; **NOTCH2**; **AGO1**; **TNRC6B**	8.7 × 10^−5^	6.2 × 10^−4^
	Signaling by TGF-beta Receptor Complex	CCNT2; **SMAD3**; **SP1**; **SMAD2**; XPO1; PARP1	2.5 × 10^−4^	0.0015
	NGF signaling via TRKA	**CDKN1B**; **CRK**; **CRKL**; **TNRC6A**; **AGO1**; **TNRC6B**; **MDM2**; **YWHAB**	0.0023	0.0074
	VEGFA-VEGFR2 Pathway	**CRK**; VEGFA; **PAK2**; **ACTG1**; **YWHAB**	0.0056	0.010
	Signaling by SCF-KIT	**CDKN1B**; **TNRC6A**; **AGO1**; **TNRC6B**; **MDM2**; **YWHAB**	0.0060	0.010
	Signaling by NGF	**CDKN1B**; **CRK**; BCL2L11; **CRKL**; **TNRC6A**; **AGO1**; **TNRC6B**; **MDM2**; **YWHAB**	0.0066	0.011
	PI3K/AKT activation	**CDKN1B**; **TNRC6A**; **AGO1**; **TNRC6B**; **MDM2**	0.0070	0.011
	DAP12 signaling	**CDKN1B**; **TNRC6A**; **AGO1**; **TNRC6B**; **MDM2**; **YWHAB**	0.012	0.013
	Signaling by EGFR	**CDKN1B**; **TNRC6A**; **AGO1**; **TNRC6B**; **MDM2**; **YWHAB**	0.018	0.018
DisGeNET	Acro-Osteolysis	**NOTCH2**; ATL3	0.0029	0.010
	Pathological fracture	**CCND1**; **NOTCH2**	0.0066	0.010

**Table 4 ijms-23-06534-t004:** Pathway enrichment using the KEGG, Reactome, and DisGeNET databases. Several pathways significantly enriched with an adjusted *p* value < 0.2 (FDR < 20%) were identified for the seven miRNAs rescued by insulin. The top targeted genes most commonly present in the identified pathways are highlighted (19 genes).

Database	Pathway	Hits	*p*-Value	FDR
KEGG	Adherens junction	**RHOA**, **SMAD4**, IGF1R, IQGAP1, PTPN1, **CTNNA1**, **ACTB**	1.5 × 10^−4^	0.0055
	Apoptosis	**AKT1**, **XIAP**, IL1R1, IL1RAP, CYCS, TNFRSF10B	0.0026	0.045
	Wnt signaling	**RHOA**, **SMAD4**, **FZD1**, NFAT5, **FZD4**, VANGL1, **LRP5**, **MYC**	0.0027	0.045
	Axon guidance	**RHOA**, NFAT5, EFNB2, ITGB1, EPHB4, ARHGEF12, SRGAP1	0.0035	0.05
	Focal adhesion	**RHOA**, **AKT1**, **XIAP**, IGF1R, ITGB1, **ACTB**, LAMC1, FLNB, PPP1CC	0.0061	0.068
	Jak-STAT signaling	SPRY4, SPRED1, **AKT1**, **MYC**, IRF9	0.025	0.14
	Tight junction	**RHOA**, **AKT1**, TJP2, **CTNNA1**, **ACTB**	0.048	0.21
Reactome	Signaling by Wnt	**RHOA**, SOX4, **FZD1**, **TNRC6B**, **AKT1**, **XIAP**, **FZD4**, USP34, **AGO2**, **CALM3**, **AGO3**, **LRP5**, **MYC**, CLTC, KMT2D	7.2 × 10^−5^	0.0040
	PI3K/AKT activation	**RHOA**, **TNRC6B**, **TRIB3**, **AKT1**, **CDKN1B**, **AGO2**, **AGO3**, **MDM2**, **RICTOR**	10^−4^	0.0040
	Signaling by NGF	**RHOA**, **TNRC6B**, **TRIB3**, **AKT1**, **CDKN1B**, BCL2L11, **AGO2**, **CALM3**, RPS6KA5, **AGO3**, **MDM2**, SQSTM1, **RICTOR**, ARHGEF12	7.8 × 10^−4^	0.0049
	Ca2+ pathway	**TNRC6B**, **FZD4**, **AGO2**, **CALM3**, **AGO3**	9.7 × 10^−4^	0.0049
	Signaling by NOTCH	NUMB, **TNRC6B**, **AGO2**, B4GALT1	0.0013	0.053
	Beta-catenin independent WNT signaling	**RHOA**, **FZD1**, **TNRC6B**, **FZD4**, **AGO2**, **CALM3**, **AGO3**, CLTC	0.0015	0.0054
	VEGFA-VEGFR2 Pathway	**RHOA**, **AKT1**, AXL, **CALM3**, **CTNNA1**, **ACTB**, **RICTOR**	0.0012	0.0063
	Signaling by FGFR	**TNRC6B**, **TRIB3**, **AKT1**, **CDKN1B**, **AGO2**, **CALM3**, **AGO3**, **MDM2**, **RICTOR**	0.00257	0.0069
	DAP12 signaling	**TNRC6B**, **TRIB3**, **AKT1**, **CDKN1B**, **AGO2**, **CALM3**, **AGO3**, **MDM2**, **RICTOR**	0.0029	0.0074
	Signaling by SCF-KIT	**TNRC6B**, **TRIB3**, **AKT1**, **CDKN1B**, **AGO2**, **AGO3**, **MDM2**, **RICTOR**	0.0042	0.0095
DisGeNET	Abnormality of the vertebral column	**LRP5**, NT5E	0.0010	0.010
	Osteopetrosis	**SNX10**, **LRP5**	0.0034	0.014
	Varying degree of multiple fractures/increased fracture rate	**FZD4**, **SNX10**, **LRP5**	0.0123	0.018
	Osteopetrosis, autosomal recessive 8	**SNX10**	0.0188	0.018
	Osteopetrosis, autosomal dominant 1/osteoporosis with pseudoglioma/bone mineral density quantitative trait locus 1	**LRP5**	0.0188	0.018

## Data Availability

The sequencing data are available in the Gene Expression Omnibus repository (GSE201231).

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
