# Peer review of "Effect of Anti-Osteoporotic Treatments on Circulating and Bone MicroRNA Patterns in Osteopenic ZDF Rats"

_ijms, 2022, doi:10.3390/ijms23126534_

Round 1

Reviewer 1 Report

This MS is well written and provides new insight to the field. Only few minor comments and suggestion. 

Figure 6: add error bar and statistics 

All rats that were used for this study were: Simplify 

starting at 11 weeks of age for 12 weeks: Sex and bodyweight?

by the local Institutional Animal Care Committee.: Approval number

 TUD (Technical University of Dresden, Germany) and used: Explain at first usage, not here.

4.1. Animals: This section needs more details like how many animals used? How many groups? how many normal and diseased? Treatment procedure before sacrifice? mode of sacrifice animal? 

4.5. Statistical Analysis: Would be better to move this section end of the Method part.

Author Response

1.    Figure 6: add error bar and statistics 
Response: We thank the reviewer for this suggestion. However, Figure 6 shows the absolute quantity (in counts per million) of the specific miRNA in a given sample (each sample shown in the x-axis). This is based on the FANTOM atlas, which gives the expression profile among many different samples for many different miRNAs analyzed in that study. Because of that, error bars and statistics would not apply to this graph, since the absolute quantity of the analyzed miRNA for a specific sample is given.
2.    All rats that were used for this study were: Simplify 
Response: We thank the reviewer for this suggestion. 
Author action: We simplified that sentence for the following: “For this study, rats were”
3.    starting at 11 weeks of age for 12 weeks: Sex and bodyweight?
Response: We thank the reviewer for this suggestion. 
Author action: We added the following sentence: “All rats were male and had an initial weight range of 350-400 g” (https://www.criver.com/products-services/find-model/zdf-rat-obese?region=3661 –> See section Baseline Colony Data-Weight Chart)
4.    by the local Institutional Animal Care Committee.: Approval number
Response: We thank the reviewer for this suggestion. 
Author action: We added the approval number: 24D-9168.11-1/2008-30
5.    TUD (Technical University of Dresden, Germany) and used: Explain at first usage, not here.
Response: We thank the reviewer for this suggestion. 
Author action: We added “(Technical University of Dresden, Germany)” when TUD acronyms was first used in the same section. 
6.    4.1. Animals: This section needs more details like how many animals used? How many groups? how many normal and diseased? Treatment procedure before sacrifice? mode of sacrifice animal? 
Response: We thank the reviewer for this suggestion. 
Author action: All of that information is provided in Table 1 within section 2.1. Study design. Only mode of sacrifice was missing and added within section 4.1. Animals.
7.    4.5. Statistical Analysis: Would be better to move this section end of the Method part.
Response: We thank the reviewer for this suggestion. 
Author action: Section 4.5. Statistical Analysis was moved to the end of the Method part.

Reviewer 2 Report

The manuscript titled “Effect of anti-osteoporotic treatments on circulating and bone microRNA patterns in osteopenic ZDF rats” is an interesting and novel study. The authors have attempted to explore possible miRNA markers for diabetic osteoporosis. This study will also help to establish new miRNA targets for diabetic osteoporosis treatment. The manuscript is well written and the data are well presented. The authors may add a brief description of the methods used for the 'Animal Phenotype analysis.' Adding the details of anti-scleorstin antibody and PTH dosing will be help full.

Author Response

  1. The authors may add a brief description of the methods used for the 'Animal Phenotype analysis.' Adding the details of anti-scleorstin antibody and PTH dosing will be help full.

Response: We thank the reviewer for this suggestion. Regarding the dosing of PTH and anti-sclerosting, that information is already given in section 2.1. Study design.

Author action: We added a brief description of the methods used to obtain the rat phenotype in section 4.1. Animals.